# Single-domain antibody inhibitors target the coiled coil arms of the *Bacillus subtilis* SMC complex

Ophélie J Gosselin[1], Michael Taschner[1], Lea M Huber-Hürlimann[2], Markus A Seeger[2], Stephan Gruber[1]*

[1]Department of Fundamental Microbiology (DMF), Faculty of Biology and Medicine (FBM), University of Lausanne (UNIL), Lausanne, Switzerland; [2]Institute of Medical Microbiology, University of Zurich, Zürich, Switzerland

## eLife Assessment

This **important** study introduces an innovative synthetic nanobody approach to probe the function of the bacterial SMC complex. The work is a **compelling** example of the potential of this approach. The authors generate protein chimeras to provide **convincing** evidence that their identified nanobodies target the coiled-coil region of the SMC subunit, demonstrating that this region is critical for SMC function in vivo. Overall, the work is significant for the fields of genome organization, SMC protein biology, synthetic biology, and bacterial cell biology.
[Editors' note: this paper was reviewed by Review Commons.]

*For correspondence:
stephan.gruber@unil.ch

Competing interest: The authors declare that no competing interests exist.

**Abstract** Synthetic nanobodies—also called sybodies—have proven valuable for stabilizing conformations of purified proteins, advancing structural and functional studies for example of transmembrane proteins. However, their utility in modulating protein function in living cells has remained less well explored. Structural Maintenance of Chromosomes (SMC) complexes facilitate chromosome organization, a fundamental process in all domains of life. In this study, we target the bacterial SMC complex, Smc-ScpAB, in *Bacillus subtilis* with synthetic nanobodies, aiming to identify key functional regions of the protein complex in a largely unbiased manner. We first isolate sybodies that specifically bind purified Smc-ScpAB and then express them in *B. subtilis* to select binders capable of disrupting Smc-ScpAB function, leading to chromosome segregation defects and cell death. Mapping and biochemical characterization show that the 14 disruptive sybodies belong to one of three library designs, target the Smc subunit near the same coiled coil arm interface and modulate its ATPase activity in two principal ways, highlighting the mid-region of the Smc coiled coil as critical feature of the SMC-DNA folding process. These findings underscore the potential of sybodies—and, by extension, designed binders—as versatile tools for probing dynamic protein function in living cells.

## Introduction

Rapid and specific interference with protein activity and dynamics in living cells is essential for studying and understanding biological mechanisms. However, current approaches—primarily based on small-molecule inhibitors—remain laborious, time-consuming, and costly. The advent of conformation-specific synthetic nanobody (sybody) selection (*Zimmermann et al., 2018*), and more recently, computational design of protein binders (*Pacesa et al., 2025*), offer unique opportunities for

protein-based inhibition and modulation of cellular targets. Here, we generate sybodies against the SMC complex in *Bacillus subtilis* to efficiently inhibit its function within cells.

Structural Maintenance of Chromosomes (SMC) complexes are multi-subunit, ring-shaped ATP-hydrolyzing DNA motors that structure chromosomal DNA by loop extrusion. They are essential for chromosome organization and segregation, the regulation of gene expression, DNA repair, and defense against non-self DNA across all domains of life (*Yatskevich et al., 2019*). Each SMC protein (Smc in *B. subtilis*, *bsu*Smc) is a long polypeptide with its N- and C-termini folding together to form a globular ATP binding cassette (ABC) 'head' domain. The head is connected to a 'hinge' dimerization domain via an ~50 nm antiparallel coiled coil 'arm', creating an elongated dimer (*Figure 1Aii and iii*). ATP binding promotes head engagement, bringing the heads together to form a functional ATPase. Non-SMC subunits, a kleisin (ScpA in *B. subtilis*) and a dimer of KITE (ScpB in *B. subtilis*) or two HAWK proteins, bridge the heads, together forming a ring around DNA (*Figure 1Aiii*; *Wilhelm et al., 2025*). In *B. subtilis*, Smc-ScpAB is recruited to the origin of replication (*oriC*) region on the chromosome by ParB, a DNA-binding protein that recognizes centromere-like *parS* sequences (*Figure 2Ai*). Once loaded, Smc-ScpAB translocates at a rate of ~1 kb/s onto flanking DNA, aligning the two chromosome arms and individualizing nascent sister chromosomes (*Bürmann and Gruber, 2015*; *Gruber and Errington, 2009*; *Wang et al., 2014*), likely bypassing obstacles on the chromosome through the SMC hinge gate (*Liu et al., 2025*). Null mutants of *smc*, *scpA*, or *scpB* fail to segregate chromosomes properly and lose viability under conditions promoting rapid growth (*Gruber and Errington, 2009*; *Wang et al., 2014*; *Sullivan et al., 2009*; *Gruber et al., 2014*).

Several models have been proposed for the mechanism of DNA loop extrusion by SMC complexes, including the 'segment capture' model, in which DNA segments are transiently trapped between the SMC arms in an open state and fused into larger loops through iterative ATP-driven cycles (*Figure 1Aiii*, *Figure 1—figure supplement 1*; *Diebold-Durand et al., 2017*; *Marko et al., 2019*; *Minnen et al., 2016*). However, detailed structural understanding and experimental testing in vivo are lacking.

Antibodies can block enzymatic reactions by stabilizing reaction intermediates. Synthetic single-domain antibodies called sybodies are small, robust antigen-binding proteins that were engineered based on three camelid nanobody structures (*Zimmermann et al., 2018*). They are selected from synthetic libraries that encode a wide diversity of binding surfaces and epitope shapes based on the length and geometry of their CDR3 loop, allowing strong binding of diverse antigen surfaces (*Figure 1Ai*). Their small size (~15 kDa), stability, and ability to bind transient epitopes make them ideal tools for targeting specific states of biomolecules (*Rasmussen et al., 2011*). Selection is carried out using purified and immobilized protein by ribosome display and phage display, typically followed by an ELISA-based screening (*Zimmermann et al., 2018*).

Here, we demonstrate that sybodies can be used to interfere with the function of *B. subtilis* Smc-ScpAB in vivo. We first isolate binders that specifically target purified Smc-ScpAB in vitro and then select those that eliminate Smc function when expressed in *B. subtilis*. Fourteen sybodies were found to disrupt chromosome segregation, mimicking *smc* deletion phenotypes. Biochemical assays showed that the selected sybodies alter the ATP hydrolysis rate of Smc-ScpAB and its stimulation by DNA addition, likely stabilizing intermediates of the conformational cycle. Mapping experiments using chimeric Smc constructs revealed that most binders, unexpectedly, target the coiled coil, specifically in a defined region near the Smc 4N arm-to-arm contact (*Vazquez Nunez et al., 2021*), highlighting the potential of sybodies and designed binders, as genetic tools to identify essential functional domains and underscoring the importance of coiled coil dynamics for Smc function.

## Results

### Generation of *bsuS*mc-ScpAB-specific sybodies

To isolate sybodies that impede the function of *B. subtilis* Smc-ScpAB complex, we first performed in vitro selection with purified Smc-ScpAB in the presence of 40 bp duplex DNA, and an ATPase-deficient mutant (E1118Q) of *bsuS*mc. These conditions favor ATP-engaged 'open' complexes alongside the typically predominant ATP-disengaged rod-shaped 'closed' state (*Vazquez Nunez et al., 2021*) (*Figure 1Aiii* ). *bsuS*mc was also biotinylated at the hinge at residue R643C for immobilization (*Figure 1Aii-iii*, *Figure 1—figure supplement 1*; *Zimmermann et al., 2018*; *Bürmann et al., 2017*; *Hirano and Hirano, 2004*; *Zimmermann et al., 2020*). Starting from three synthetic sybody libraries

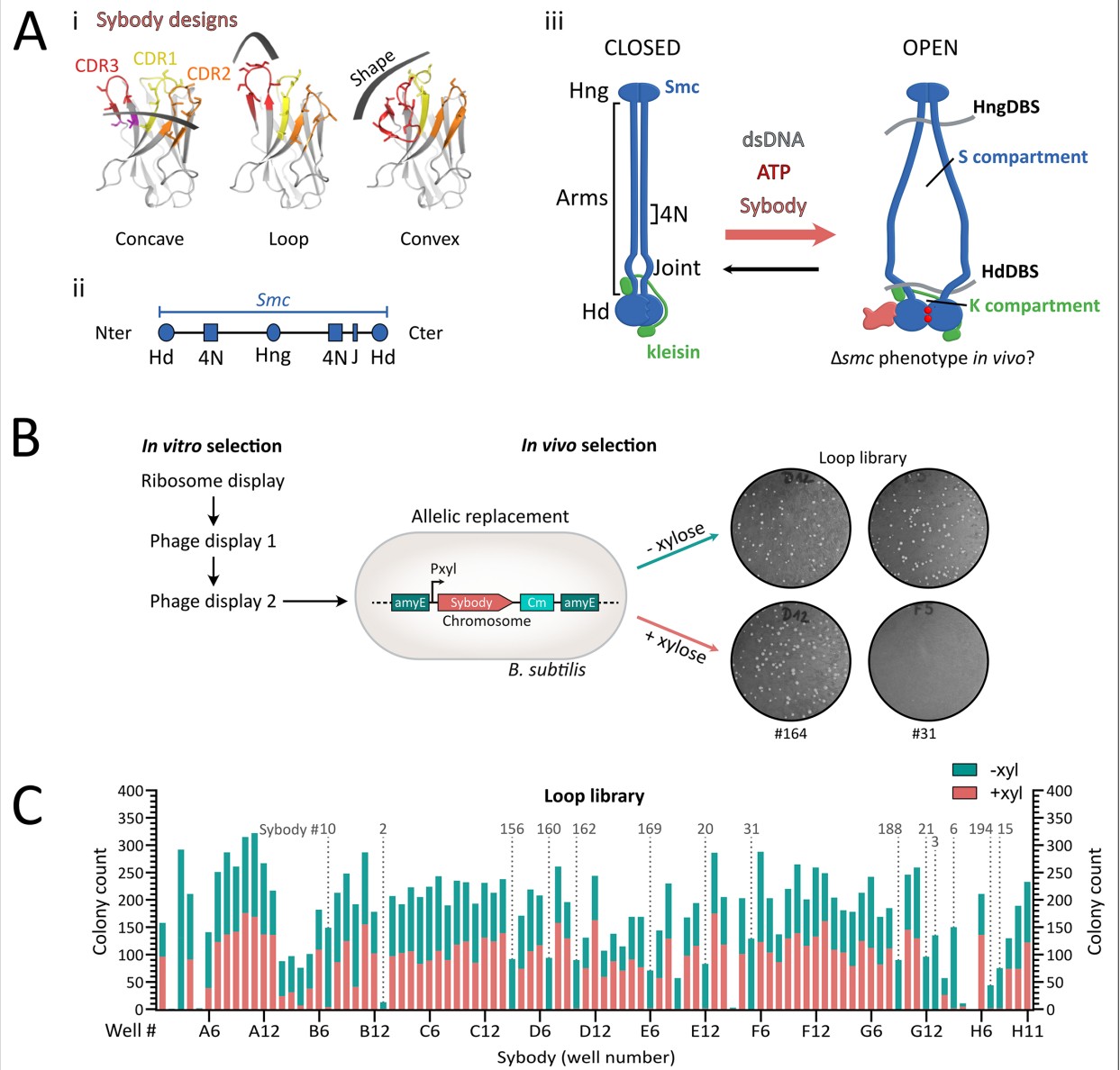

**Figure 1.** Selection of sybodies targeting Smc-ScpAB in *B. subtilis*. (**A**) (**i**) Structural design and randomization scheme of the three synthetic sybody libraries: concave, loop, and convex. Complementarity-determining regions (CDRs) 1, 2, and 3 are shown in yellow, orange, and red, respectively; randomized residues are shown in stick representation. Adapted from *Zimmermann et al., 2018*. (**ii**) SMC domain organization. Hd: head, Hng: hinge, 4N: 4N arm-to-arm contact, J: joint. (**iii**) SMC complexes harbor juxtaposed arms in their 'closed' state (left panel). Upon ATP binding at the heads, the heads engage, leading to an 'open' state in which the arms disengage (right panel). This state is thought to accomodate a segment of DNA (not shown) in the S compartment formed by the SMC arms, that will be pushed towards the K compartment formed by the heads and kleisin, after ADP release. Blue: SMC dimer, Green: kleisin ScpA. The ScpB KITE subunits are not represented for the sake of simplicity. HngDBS: Hinge DNA-binding site, HdDBS: Head DNA-binding site. (**B**) Framework for sybody selection. In vitro selection starts with ~$10^{12}$ sybody variants per library subjected to ribosome display for pre-enrichment, followed by two rounds of phage display. For in vivo selection, 95 randomly selected sybody genes were integrated into the *B. subtilis* chromosome under the control of a xylose-inducible promoter by allelic replacement at the *amyE* locus. Growth defects on rich medium were tested on ONA agar plates with or without 0.5% xylose. Example shown: Sb164 (loop library) did not affect growth, whereas Nb31 impaired growth upon induction, suggesting interference with *bsuSmc* function. (**C**) Transformation assay results for a *B. subtilis* straintransformed with one out of 95 sybodies of the loop library expressed from a xylose-inducible promoter . Bars show colony counts 'without' (green) on top of 'with' (pink) xylose. Strains are ordered by their original position in the 96-well plate. Fourteen sybodies consistently impaired colony formation under inducing conditions (marked by dotted lines). Sybody numbers indicated above the plots correspond to selected candidates used in subsequent experiments, numbering according to order of first use. Notably, the E09 sybody (Sb018) also showed an absence of transformants upon sybody induction. However, this sybody candidate gave intermediate phenotypes in later experiments, which is why it was excluded from detailed analysis.

*Figure 1 continued on next page*

*Figure 1 continued*

The online version of this article includes the following source data and figure supplement(s) for figure 1:

**Figure supplement 1.** Preparation of *bsuS*mc(C119S, C437S, C826S, E1118Q, R643C)-ScpAB complex and loop extrusion model.

**Figure supplement 1—source data 1.** Original TIF image file of protein gel shown in *Figure 1—figure supplement 1A*.

**Figure supplement 1—source data 2.** PDF file containing image of protein gel shown in panel *Figure 1—figure supplement 1A* with lanes and bands labeled.

**Figure supplement 2.** Representative results from in vivo sybody selection based on colony formation.

encoding distinct epitope-binding geometries (denoted as concave, loop, convex, respectively, *Figure 1Ai*), we performed one round of ribosome display starting from a large sybody library followed by two rounds of phage display, thereby enriching sybodies that bind to Smc-ScpAB (*Table 1*). From each library, 95 sybody-expressing *Escherichia coli* clones were randomly chosen for further characterization. ELISA data revealed that nearly all clones bind purified Smc-ScpAB (*Table 1*). However, the ELISA signals of only few Sybodies showed clear dependence on the presence or absence of ATP and DNA (*Supplementary file 1*).

## Identification of *bsuS*mc-ScpAB-blocking sybodies through phenotypic screening in *B. subtilis*

We next screened for inhibitory sybodies directly in *B. subtilis* cells. Despite containing a conserved disulfide bond, sybodies have been successfully expressed inside cells and were shown to bind their targets in the disulfide reducing environment of the cytoplasm, likely owing to their robust folding (*Deneka et al., 2021*). To assess whether the selected sybodies interfere with Smc function in vivo, each of the 285 sybody sequences was cloned under a xylose-inducible promoter ($P_{xyl}$) in an *E. coli–B. subtilis* shuttle vector and integrated at the *amyE* locus of the wild-type *B. subtilis* 1A700 strain (*Figure 1B*; *Diebold-Durand et al., 2019*). Fourteen sybodies from the loop library failed to yield transformants in the presence (but not absence) of xylose, indicating cell lethality due to sybody inhibition of Smc-ScpAB (*Figure 1C*, *Figure 1—figure supplement 2*). These Smc-disruptive sybodies (denoted as Sb002, 003, 006, 010, 015, 020, 021, 031, 156, 160, 162, 169, 188, and 194, respectively) harbor 14 distinct sequences (*Supplementary file 2*). Intriguingly, no such disruptive sybodies were isolated from the concave and convex libraries (*Figure 1—figure supplement 2*), suggesting that the CDR3 geometry as present in the loop library is particularly effective at inhibiting *bsuS*mc-ScpAB. Likely, sybodies of the other libraries bound to other epitopes of *bsuS*mc-ScpAB that are less sensitive to conformational trapping, and it is further possible that they were not stably expressed in *B. subtilis*. However, at least for sybodies from the loop library, we did not notice any obvious correlation between expression levels and phenotypes. Selected sybodies showed similar expression and were appreciably expressed even without inducer (*Figure 2—figure supplements 2 and 3*). Notably, tendencies of preferential isolation of binders from one of the three libraries have previously been observed, although at milder levels (*Zimmermann et al., 2018*; *Deneka et al., 2021*; *Hutter et al., 2019*).

## Selected sybodies impede chromosome segregation

To determine whether the 14 sybodies indeed impair chromosome segregation, we used fluorescence microscopy to monitor *oriC* positioning in a *B. subtilis* strain expressing a ParB-GFP fusion protein together with a $P_{xyl}$-inducible sybody gene. In these strains, ParB-GFP marks the replication origin region by binding to *parS* sites near *oriC* (*Figure 2Ai*). As expected, the Δ*smc* strain displayed fewer ParB-GFP foci and elongated cells (*Figure 2Ai*; *Supplementary file 3*). These elongated cells are known to harbor expanded nucleoids, consistent with delayed *oriC* separation rather than delayed DNA replication (*Wang et al., 2014*; *Gruber et al., 2014*). A time course experiment using sybody Sb006 revealed chromosome organization defects as early as 30 minutes post induction, with a significant reduction in ParB-GFP foci per μm cell length at 40 minutes (p=0.0145) (*Figure 2Aii*). Subsequent imaging was performed ~35 minutes after induction of sybody expression.

All disruptive sybodies reduced ParB-GFP foci density compared to the control (1.03 foci/μm), with values ranging from 0.56 foci/μm (Sb020) to 0.22 foci/μm (Sb003). These defects were all robustly detected, while being less severe than the Δ*smc* mutant (~0.05 foci/μm), conceivably due to the

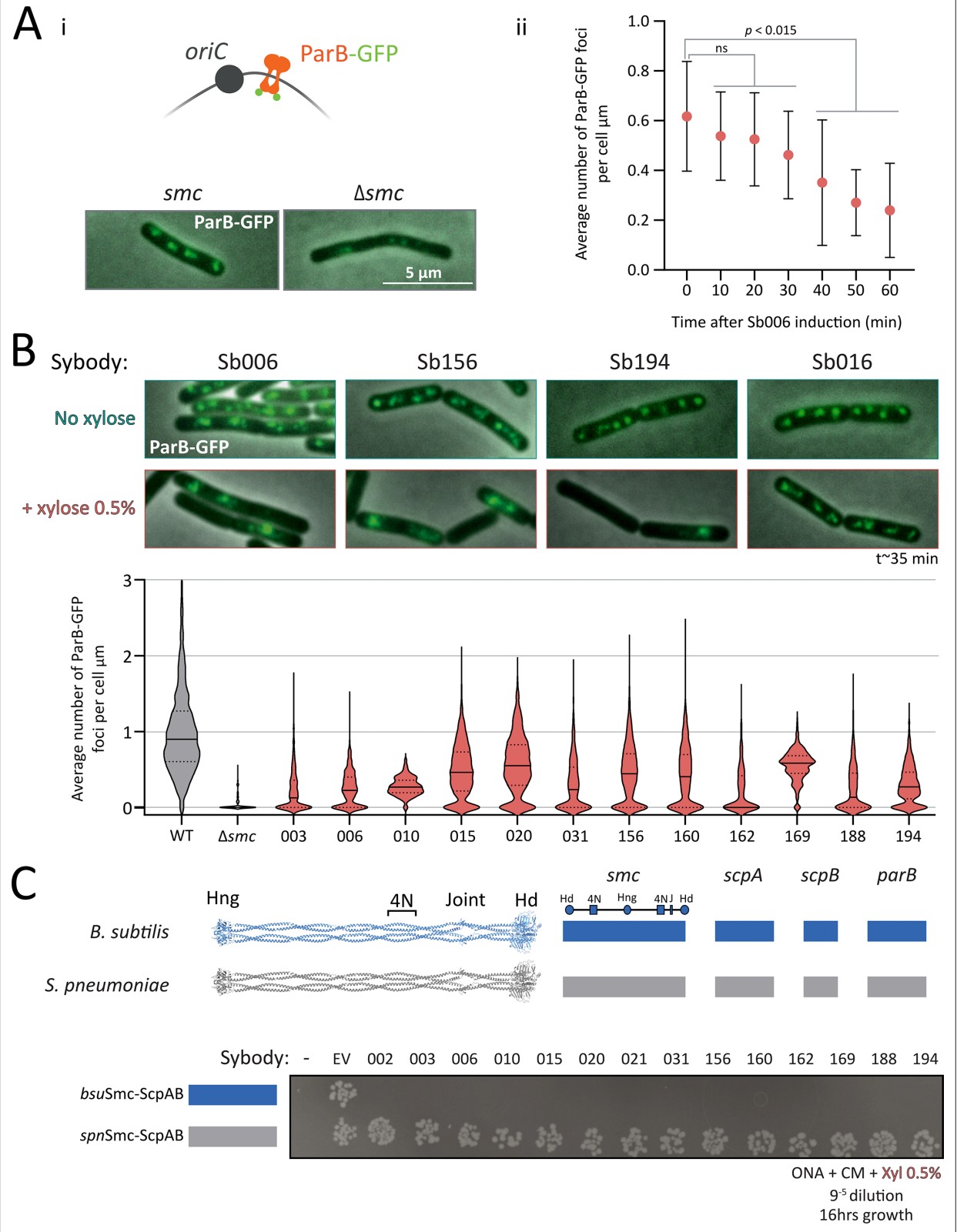

**Figure 2.** Sybody-induced chromosome segregation defects visualized by ParB-GFP imaging. (**A**) (**i**) *Top*: schematic illustrating ParB-GFP binding near *oriC*, enabling visualization of origin positioning. *Bottom*: representative images showing ParB-GFP foci in wild-type ('WT') and Δ*smc B. subtilis.* WT cells typically display 2–4 foci per cell, whereas Δ*smc* cells exhibit reduced foci numbers. (**ii**) Number of ParB-GFP foci per μm of cell length in a strain carrying inducible sybody Sb006. A significant decrease in foci is detected from 40 minutes post-induction ($p_{t0-10}$=0.9674; $p_{t0-20}$=0.9033; $p_{t0-30}$=0.3744;

*Figure 2 continued on next page*

*Figure 2 continued*

$p_{t0-40}$=0.0145; $p_{t0-50}$=0.0004; $p_{t0-60}$<0.001). Based on this, a standardized induction time of ~35 minutes was used in subsequent experiments. (**B**) ParB-GFP foci density (foci/µm) in WT, Δ*smc,* and sybody-expressing strains after 35 minutes of xylose induction. Violin plots show distribution per condition; solid lines denote the mean, and dotted lines indicate quartiles. Several hundreds of cells were analyzed (between 351 and 1735), except for Δ*smc* and Sb010 were fewer cells were available (134 and 73, respectively). (**C**) Spot assay to assess colony formation of *B. subtilis* strains harboring *bsu* or *spn* variants of Smc-ScpAB and ParB. *Top*: Schematic shows gene origin (blue: *B. subtilis*; gray: *S. pneumoniae*). The leftmost column corresponds to the parental *B. subtilis* strains without vector integration lacking chloramphenicol ('Cm') resistance (non-growing). The next spots represent the same strains carrying the Cm resistance but lakcing a sybody gene (EV for empty vector). Remaining columns are sybody-expressing strains; sybody numbers are indicated . Cells were grown for 16 hours at 37°C on ONA supplemented with 0.5% xylose and chloramphenicol. Hd: head, Hng: hinge, 4N: 4N arm-to-arm contact, J: joint.

The online version of this article includes the following figure supplement(s) for figure 2:

**Figure supplement 1.** Functional impact of sybody expression on chromosome organization, cell length, and growth in *B. subtilis*.

**Figure supplement 2.** Sybody-GFP expression at different inducer concentrations.

**Figure supplement 3.** Imaging of various Sybody-GFP proteins without inducer.

short induction time or incomplete inhibition. Milder defects were also observed without induction, likely attributed to leaky expression from the $P_{xyl}$ promoter (***Figure 2—figure supplement 1***). Sybody expression also caused cell elongation and growth delays, both hallmarks of impaired Smc activity, consistent with chromosome segregation defects that delay cell division (***Figure 2—figure supplement 1***). Altogether, these results show that all selected sybodies induce Δ*smc*-like phenotypes likely by interfering with *bsu*Smc-ScpAB activity in vivo.

## Selected sybodies specifically target *B. subtilis* Smc-ScpAB

To test whether sybodies specifically target *B. subtilis* Smc-ScpAB, we assessed their effects in a strain expressing *Streptococcus pneumoniae* (*Spn*) Smc-ScpAB and ParB proteins in place of the endogenous *B. subtilis* proteins. The *Spn* sequences together can functionally replace the corresponding *Bsu* sequences despite significant sequence divergence (~38% sequence identity) (***Bock et al., 2022***). Strikingly, none of the sybodies impaired growth in this background, even though a cognate ParAB pair is absent in this strain (notably *S. pneumoniae* naturally lacks *parA*), sensitizing cells to chromosome segregation defects (***Figure 2C***). These results confirm that the sybodies are specific to *bsu*Smc-ScpAB and that off-target toxicity is not noticeable.

## *bsuS*mc-ScpAB-disrupting sybodies target two distinct coiled coil regions adjacent to the *bsuS*mc 4N arm-to-arm contact

To map the sybody-binding site on the *bsuS*mc-ScpAB complex, we utilized five chimeric Smc constructs (Smc Chimera 1–5), in which the hinge and progressively longer segments of the adjacent coiled coils were replaced with the corresponding sequences from the *S. pneumoniae* Smc protein (***Figure 3A***). Chimeric junctions were designed to preserve coiled coil integrity based on available crystal structures, coiled coil predictions by DeepCoil, and AlphaFold2 structural models (***Gabler et al., 2020***; ***Jumper et al., 2021***). All chimeric Smc strains retained the native *B. subtilis scpA*, *scpB*, and *parB* genes and were viable under conditions promoting fast growth, demonstrating proper protein folding and functioning of chimeric Smc-ScpAB complexes (***Figure 3—figure supplement 1***).

**Table 1.** Sybody enrichments at different steps of the selection procedure.
Ribosome display output was quantified by qPCR, while phage display results include final phage titers and enrichment values from rounds one and two, also measured by qPCR.

| Library | Ribosome display (qPCR) | Phage display (qPCR) | | |
|---|---|---|---|---|
| | Total # of RNAs | Titer (PFU/mL) | Enrichment 1 | Enrichment 2 |
| Convex (S) | $1.36 \times 10^8$ | $8.96 \times 10^{13}$ | 3.1× | 1474.7× |
| Loop (M) | $6.95 \times 10^7$ | $5.46 \times 10^{13}$ | 1.4× | 1209.9× |
| Concave (L) | $1.25 \times 10^8$ | $8.32 \times 10^{13}$ | 1.8× | 1808.0× |

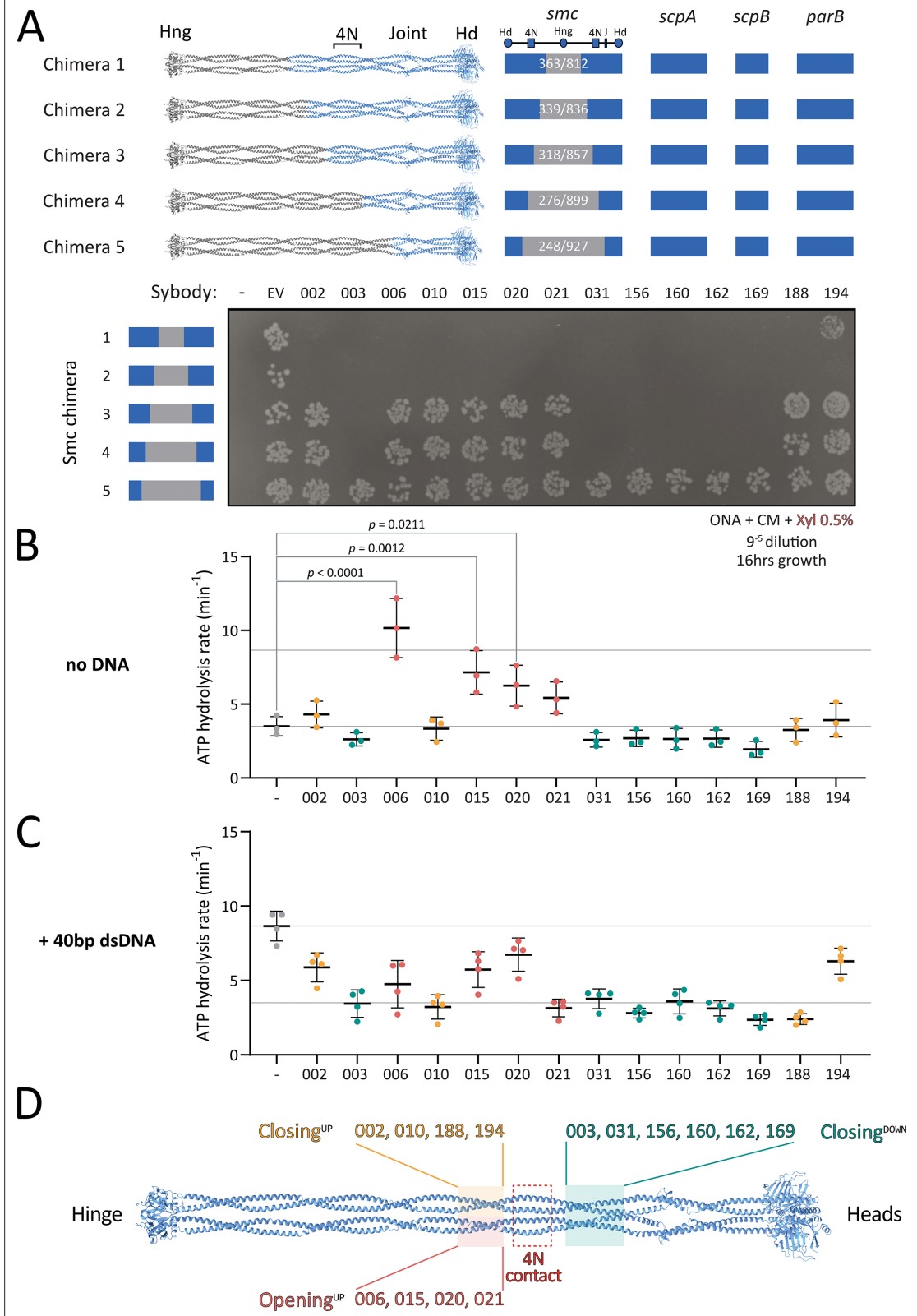

**Figure 3.** Mapping sybody binding sites on Smc-ScpAB. (**A**) Colony formation of *B. subtilis* strains harboring chimeric Smc proteins comprising *S. pneumoniae* and *B. subtilis* sequences and expressing sybodies The schematic above the spot assay depicts the species origin of various parts of the *smc* gene (blue: *B. subtilis*; gray: *S. pneumoniae*). '–' indicates no insertion at the *amyE* locus; 'EV' refers to the empty vector control containing only the chloramphenicol resistance cassette at *amyE*; numbered labels correspond to sybodies. Cells were spotted on nutrient rich medium (ONA)

*Figure 3 continued on next page*

*Figure 3 continued*

supplemented with chloramphenicol and xylose and incubated for 16 hours at 37°C. Hd: head, Hng: hinge, 4N: 4N arm-to-arm contact, J: joint. (**B**) ATP hydrolysis rates of *bsu*Smc-ScpAB in the presence of sybodies but absence of DNA. Means and standard deviation from three technical replicates are indicated. Individual datapoints are shown. Significant effects by one-way ANOVA are indicated by p values. (**C**) ATPase rates in the presence of 40 bp dsDNA. All sybodies reduced DNA-stimulated ATP hydrolysis. Means and standard deviation from four technical replicates are indicated. Individual datapoints are shown. Reported p-values: Sb020 (p=0.340), Sb194 (p=0.0049), Sb002 (p=0.0007), Sb015 (p=0.0003); all others, p<0.0001. (**D**) Schematic summary of sybody binding sites mapped onto the Smc dimer, categorized by their effect on ATPase activity. Sybodies are grouped based on functional impact and mapped to corresponding structural regions: pink/yellow boxes indicate residues 318–339 and 836–857; green boxes mark residues 248–276 and 899–927; and the red box highlights the 4N contact region (approx. residues 290–320).

The online version of this article includes the following figure supplement(s) for figure 3:

**Figure supplement 1.** Sybody effects on cellular viability and SMC ATPase rate.

Chimeras 1 and 2, which carry the central hinge and coiled coil sequences of *Spn* origin (residues 363–812 and 339–836, respectively), showed the same sybody-induced growth defects as the wild-type strain, indicating that the sybodies bind further away from the hinge, within the remaining *B. subtilis* portion comprising the Smc head and the first ~230 coiled coil residues, and ScpAB (*Figure 3A*). By contrast, Chimera 5 (residues 248–927 of *Spn* origin) supported robust growth even in the presence of the *bsu*Smc-disrupting sybodies, indicating that the region comprising the heads and first ~50 residues of the coiled coil (including the joint) does not comprise the binding site. The binding region for the sybodies is thus located on the coiled coil between residues 248 and 339 (or 836 and 927), demonstrating that all 14 sybodies bind to a central region of the coiled coil.

Chimera 3 (*spn*Smc residues 318–857) and chimera 4 (*spn*Smc residues 276–899) revealed two distinct sybody responses. Sb003, 031, 156, 160, 162, and 169 impaired growth of both strains, mapping their binding sites to *B. subtilis* residues 248–276 (and 899–927), present in chimeras 1–4 but absent in chimera 5. In contrast, Sb002, 006, 010, 015, 020, 021, 188, and 194 had no impact on the viability of chimera 3 or 4, suggesting they target residues 318-339/836-857 (*Figure 3A and D*).

In sum, these experiments mapped the binding sites of all 14 disruptive sybodies to one of two ~twenty amino acid segments in the central region of the *bsu*Smc coiled coil (*Figure 3D*). These regions harbor relatively poorly conserved sequences but flank the 4N arm-arm contact (~at residue 295), previously proposed to serve as an essential structural switch between closed and open conformations (*Vazquez Nunez et al., 2021*). These results reveal that all isolated sybodies target the same functional element of the Smc arms, and that interfering with this element disrupts Smc-dependent chromosome organization in vivo.

## Smc-disrupting sybodies affect the ATPase activity in one of two ways

To investigate how sybodies may influence *bsu*Smc-ScpAB, we next measured ATP hydrolysis rates using purified proteins in the presence and absence of DNA. SMC ATPase activity serves as an indirect readout for state transitions, including arm opening and head engagement. In absence of DNA, *bsu*Smc-ScpAB hydrolyzes ATP at ~4 ATP/min/Smc (*Vazquez Nunez et al., 2021*). DNA binding stimulates ATP hydrolysis to ~9 ATP/min/Smc, presumably by DNA binding promoting arm opening and head engagement, creating a more open conformation.

All disruptive sybodies measurably affected the ATP hydrolysis rate of *bsu*Smc-ScpAB. Ten maintained a near-basal ATPase rate even in presence of DNA (Sb002, Sb003, Sb010, Sb031, Sb156, Sb160, Sb162, Sb169, Sb188, and Sb194) (*Figure 3B and C*, *Figure 3—figure supplement 1*). This suggests that they stabilize the complex in a closed or partially closed conformation, thus hindering DNA-dependent ATPase stimulation. Notably, in this group, Sb002 and Sb194 showed slightly elevated ATPase rates with DNA, hinting at a mildly more flexible conformation.

In contrast, the other sybodies (Sb006, Sb015, Sb020, and Sb021) stimulated ATP hydrolysis in the absence of DNA, with Sb006 reaching levels comparable to DNA-induced activation (~10 ATP/min). This implies that these sybodies stabilize Smc-ScpAB in a more open state promoting head engagement. Curiously, addition of DNA in the presence of Sb006 and Sb021 reduced—rather than stimulated—ATPase activity (*Figure 3B and C*), suggesting that concurrent DNA and sybody binding may trap the complex in a non-productive conformation, perhaps by favoring the open state excessively. These findings are consistent with the idea that 4N contact region plays a critical role in arm opening and ATPase stimulation.

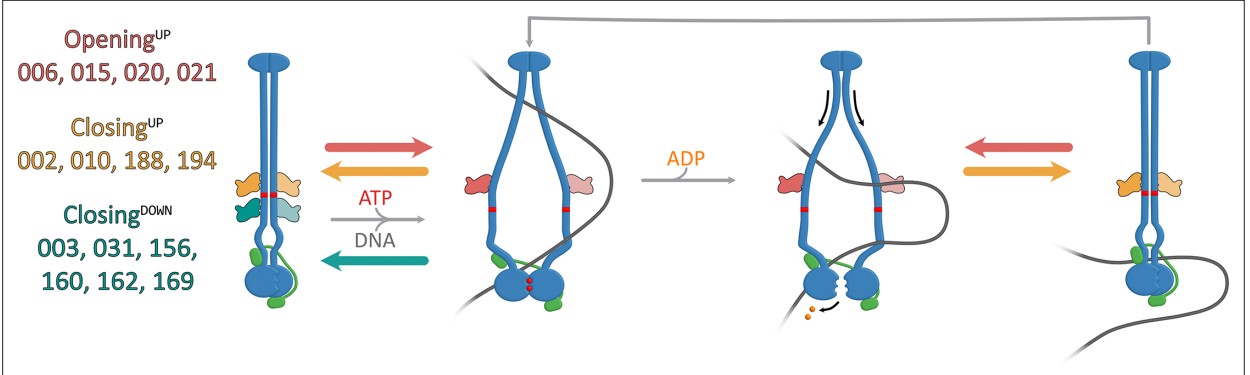

**Figure 4.** Proposed models for sybody interactions with *bsu*Smc-ScpAB . Model for sybodies of the Opening[UP], Closing[UP], and Closing[DOWN] group. Opening[UP] sybodies likely prevent complete arm closure. In the presence of DNA, these sybodies may stabilize an 'open' conformation and impede head disengagement, resulting in reduced ATPase activity (not shown here). Closing[UP] and Closing[DOWN] sybodies stabilize a closed-arm conformation. These potentially hinder DNA segments from entering the inter-arm space and accessing the hinge-proximal DNA binding site.

The online version of this article includes the following figure supplement(s) for figure 4:

**Figure supplement 1.** Hypothetical model for an Opening[DOWN] class of sybodies, not recovered in this study.

Interestingly, the observed ATPase profiles correlate with distinct binding regions along the Smc arms. All ATPase-stimulating sybodies (Sb006, Sb015, Sb020, Sb021) bind above the 4N contact ('Opening[UP]'), while sybodies that hinder DNA stimulation fall into two subgroups: those binding above the 4N contact (Sb002, Sb010, Sb188, Sb194—'Closing[UP]') and those binding below it (Sb003, Sb031, Sb156, Sb160, Sb162, Sb169—'Closing[DOWN]') (*Figure 3D*). These results indicate that the sybodies perturb *bsu*Smc-ScpAB in two principal ways: either by favoring the opening of the arms or by constraining the arms in a more closed state. The results highlight arm opening and closure as central regulatory features of bacterial Smc-ScpAB (*Diebold-Durand et al., 2017*; *Minnen et al., 2016*; *Soh et al., 2015*; *Vazquez Nunez et al., 2019*) and confirm the key role of the 4N arm contact in balancing the opening and closing of SMC arms (*Vazquez Nunez et al., 2021*).

## Discussion

Over the past three decades, nanobodies and their synthetic counterparts have transformed biomedical research and structural biology by enabling the targeting and stabilization of transient protein conformations. Despite these advances, their application to probing protein function in living cells has remained limited (*Deneka et al., 2021*; *Seeger et al., 2012*). Here, we expand the utility of sybody selection by employing it as a tool to screen for genetic probes targeting the function of selected molecular machines in vivo. This strategy combines the advantages of controlled biochemical reconstitution with the ability to study the function of proteins in their native context. Importantly, sybody selection requires no prior knowledge of vulnerable regions, providing an unbiased means to interrogate protein function that is difficult to achieve through rational design. Our study based on the *B. subtilis* Smc-ScpAB DNA motor as a model uncovers a central regulatory region in the coiled coil arms of Smc and highlights how distinct arm conformations shape SMC function in loop extrusion and chromosome segregation (*Figure 4*).

### Revealing the importance of Smc arm dynamics through synthetic binders

All 14 Smc-disrupting sybodies share the loop library design. Moreover, their Smc binding sites map to the two short coiled coil segments flanking the conserved 4N arm-to-arm contact, a region that has previously been implicated in key conformational transitions during arm opening (*Vazquez Nunez et al., 2021*). While the 14 sybodies harbor different epitope binding sequences, we suspect that they may all bind to the Smc dimer in an analogous manner. With the long CDR3 loops known to bind narrow structural cavities (*Rasmussen et al., 2011*; *Kruse et al., 2013*), we speculate that these sybodies may intercalate between the two coiled coil arms (*Figure 3D*). Structural studies are needed

to test this hypothesis directly; however, our attempts at X-ray crystallography and electron microscopy of Smc-sybody complexes have failed so far.

Notably, while the coiled coil dimer appears similarly structured from the elbow to the hinge (*Diebold-Durand et al., 2017*; *Figure 3D*), no Smc-disrupting sybodies mapped to regions further away from the 4N contact (*Figure 3D*). This indicates that opening and closure of the Smc arms might be dispensable at these places, although formal testing would require isolating binders efficiently targeting these regions, potentially facilitated by binder design and expression in *B. subtilis*. This parallels with recent findings with the bacterial SMC defense system Wadjet, where hinge-proximal arm opening is dispensable for loop extrusion, but essential for obstacle bypass (*Liu et al., 2025*). Alternatively, these arm-to-arm contacts might be more stable and thus less likely to be interfered with by sybody binding. Similarly, sybodies targeting below the 4N contact all reduce the ATPase rate (or rather its DNA stimulation), indicating that artificial opening at this position may not disrupt Smc function (*Figure 4—figure supplement 1*). Again, these hypotheses need to be tested more directly. We envision that the design of protein binders may allow us to confirm these observations and further dissect the underlying mechanisms.

This work establishes sybodies as precision tools for blocking ATP-driven machines inside bacterial cells. By targeting allosteric control points in Smc, they enable mechanistic dissection of loop extrusion and open new avenues for studying dynamic complexes in vivo. More broadly, the study demonstrates how synthetic binders can trap or block conformations of active chromatin-associated machines, providing a powerful means to probe their mechanisms in living cells. Looking ahead, the rational design of protein binders with tailored geometry and allosteric potential could allow researchers to manipulate and visualize specific conformational states with even greater control and refinement, across both prokaryotic and eukaryotic organisms, while the generation and in vivo screening of sybodies remains an attractive approach to gain unexpected biological insights in an unbiased manner.

# Materials and methods

**Key resources table**

| Reagent type (species) or resource | Designation | Source or reference | Identifiers | Additional information |
|---|---|---|---|---|
| Strain, strain background (*Escherichia coli*) | *E. coli* SS320 | Lucigen | Cat# 60512-1 | For phage display |
| Strain, strain background (*E. coli*) | *E. coli* MC1061 | Lucigen | Cat# 60514-1 | Protein production |
| Strain, strain background (*E. coli*) | *E. coli* BL21-Gold (DE3) | Novagen | Cat# 69450-3 | Protein production |
| Strain, strain background (*Bacillus subtilis*) | *B. subtilis* strain 168 | Bacillus Genetic Stock Center | 1A700 | BSG1001, *trpC2* |
| Recombinant DNA reagent | M13KO7 helper phage | New England Biolabs | Cat# N0315S | For phage production |
| Recombinant DNA reagent | Plasmid pSBinit | AddGene | RRID:Addgene_110100 | Vector for initial sybody expression |
| Recombinant DNA reagent | Plasmid pET-28a(+) | AddGene | RRID:Addgene_141289 | Vector for protein expression |
| Peptide, recombinant protein | Neutravidin | Thermo Scientific | Cat# 31000 | For plate coating |
| Commercial assay or kit | Dynabeads MyOne Streptavidin T1 | Invitrogen | Cat# 65601 | Target immobilization |
| Commercial assay or kit | Dynabeads MyOne Streptavidin C1 | Invitrogen | Cat# 65001 | Target immobilization |
| Commercial assay or kit | HiTrap Blue HP 5 mL | Cytiva | Cat# 17041301 | Affinity chromatography |
| Commercial assay or kit | HiTrap Heparin HP 5 mL | Cytiva | Cat# 17040701 | Affinity chromatography |
| Commercial assay or kit | HiTrap Butyl HP 5 mL | Cytiva | Cat# 28411005 | Hydrophobic interaction chromatography |

*Continued on next page*

*Continued*

| Reagent type (species) or resource | Designation | Source or reference | Identifiers | Additional information |
|---|---|---|---|---|
| Commercial assay or kit | HiTrap Q HP 5 mL | Cytiva | Cat# 17115401 | Ion exchange chromatography |
| Commercial assay or kit | Superose 6 10/300 Increase | Cytiva | Cat# 29091596 | Size-exclusion chromatography |
| Commercial assay or kit | Superose 6 PG XK 16/70 | Cytiva | Cat# 90100042 | Size-exclusion chromatography |
| Commercial assay or kit | HiLoad Superdex 75 PG 16/600 | Cytiva | Cat# 28-9893-33 | Size-exclusion chromatography |
| Commercial assay or kit | Sepax SRT-10C SEC100 c | Sepax Technologies | Cat# 239100-10030 | Size-exclusion chromatography |
| Chemical compound, drug | EZ-Link Maleimide-PEG2-Biotin | Thermo Scientific | Cat# A39261 | Protein biotinylation |
| Chemical compound, drug | Tris(2-carboxyethyl) Phosphine Hydrochloride (TCEP) | Sigma-Aldrich | Cat# 646547 | Reducing agent |
| Commercial assay or kit | Zeba spin 7K MWCO 0.5 mL | Thermo Scientific | Cat# 89882 | Protein desalting spin columns |
| Commercial assay or kit | Nunc Maxisorp 96-well immunoplates | Merck | Cat# M9410 | Immobilization of sybodies for ELISA |
| Peptide, recombinant protein | Protein A from *S. aureus* | Merck | Cat# P3838 | Immobilization of sybodies for ELISA |
| Antibody | Mouse monoclonal anti-myc | Sigma-Aldrich | Cat# M4439; RRID:AB_439694 | Immobilization of sybodies for ELISA; 100 µL of a 1:2000 dilution |
| Commercial assay or kit | His MultiTrap HP | Cytiva | Cat# 28400989 | Affinity chromatography |
| Peptide, recombinant protein | Pyruvate Kinase/Lactate Dehydrogenase | Sigma-Aldrich | Cat# P0294-5ML | Enzyme-coupled ATPase measurement |
| Chemical compound, drug | Phosphoenol-pyruvic acid | Sigma-Aldrich | Cat# P7002-100MG | Enzyme substrate |
| Chemical compound, drug | Nicotinamide adenine dinucleotide hydrate (NADH) | Santa Cruz | Cat# 205762A | Enzyme substrate |
| Software | GraphPad Prism | GraphPad | RRID:SCR_002798 | Scientific graphing and curve fitting |

### *Bacillus subtilis* strain construction and growth

*B. subtilis* strains used in this study were derived from the parental strain 1A700. Allelic replacement was achieved via double-crossover recombination at the endogenous *smc* locus using the natural competence method described by *Diebold-Durand et al., 2019*; *Bürmann et al., 2013*. Chromosome-integration of sybody genes were performed by transforming a vector containing an *amyE* homologous regions on each side of the recombinant gene of interest, as described in *Diebold-Durand et al., 2019*. Transformants were selected on ONA solid medium containing the appropriate antibiotics. Following transformation, strains were purified by single colony isolation and verified through a combination of marker testing, phenotype assessment, PCR, and Sanger sequencing of the *smc* locus, when appropriate. For spot assays, cells were cultured to stationary phase in liquid LB medium, and $9^{-2}$ and $9^{-5}$ dilutions were then spotted onto solid ONA medium with chloramphenicol (5 ug/mL) and xylose (0.5% w/v) when induction of the $P_{xyl}$ promoter was required. The strains, plasmids, and oligonucleotides used in this study are listed in *Supplementary file 4*, respectively.

For viability assessment, 200 LB medium was inoculated with a 1:1000 dilution of a dense *B. subtilis* culture and grown to stationary phase. The culture was then serially diluted in LB medium. From each dilution ($9^{-1}$ to $9^{-8}$), 5 µL was spotted onto LB agar plates containing the appropriate antibiotic selection. Plates were incubated, and viability was assessed by imaging the spots at 16- and 24 hours post-incubation.

### Sybody selection

Sybody selection was performed following the protocol by *Zimmermann et al., 2020*, with modifications to include binders specific for the ATP/DNA-stabilized 'open' form of the *bsuSmc*-ScpAB complex.

## Ribosome display

The standard WTB buffer (50 mM Tris/acetate pH 7.4, 150 mM NaCl, 50 mM MgAc$_2$) and its derivatives were adjusted to 50 mM NaCl to facilitate *bsuS*mc-ScpAB binding to dsDNA. The target complex was assembled by mixing 500 nM *bsuS*mc(C119S, C437S, C826S, E1118Q, R643C)-ScpAB, 2 mM ATP, and 5 μM 40 bp dsDNA in WTB-D-BSA buffer (WTB including 0.5% [w/v] BSA and 0.1% Tween-20), incubated for 15 minutes at room temperature (RT). The ribosome display panning solution was supplemented with 2 mM ATP and 5 μM dsDNA to maintain binding conditions. Solution panning and biotinylated target capture were conducted at RT to optimize selection for the 'open' complex. Washing steps incorporated 2 mM ATP and 5 μm 40 bp dsDNA, while elution, reverse transcription, and cDNA amplification followed the original protocol. Reverse transcribed RNA molecules were quantified by qPCR.

## First round of phage display

Phagemid libraries were cloned and electroporated into *E. coli* SS320 as previously described (*Zimmermann et al., 2020*). Phage production was performed using M13KO7 helper phage. The phage display buffer (TBSM) consisted of 20 mM Tris-HCl (pH 7.4), 50 mM NaCl, and 2 mM MgCl$_2$. Target preparation involved incubating 500 nM biotinylated *bsuS*mc(C119S, C437S, C826S, E1118Q, R643C)ScpAB with 2 mM ATP and 5 μM dsDNA in TBSM-BSA-D (TBSM including 0.5% [w/v] BSA and 0.05% Tween-20). Phages (10$^{12}$/mL) were incubated with the target at 50 nM for 20 minutes at RT before immobilization on neutravidin-coated plates. Washing (with added 2 mM ATP and 5 μm 40 bp dsDNA) and elution otherwise followed the original protocol, with enrichment assessed by qPCR. Amplified phages were used for a second selection round.

## Second round of phage display

Purified phages from the first round were used at 5×10$^{13}$ phages/mL, and the target was prepared as in the first round. Phage-target binding occurred at 50 nM in the presence of ATP and dsDNA, followed by capture on magnetic beads. A competition step using non-biotinylated *bsuS*mc-ScpAB (1 μM) was included before washes with TBSM-D. Phages were eluted and enrichment was determined by qPCR. Phagemids were purified, sub-cloned into the pNb_init vector, and transformed into *E. coli* MC1061.

## Growth curve analyses

Growth curves were generated from *B. subtilis* strains grown overnight to exponential phase in LB medium supplemented with 0.5% glucose, incubated at 30°C with shaking. The following day, fresh 5 mL LB cultures were inoculated with the overnight culture to an OD of 0.005 and incubated at 37°C with shaking until reaching an optical density (OD) of 0.05. Subsequently, each culture was subjected to a twofold dilution in a 96-well plate (Costar #3596), achieving a final volume of 200 μL per well. Where necessary, cultures were induced with 0.5% xylose. Plates were incubated at 37°C with continuous shaking in a Thermo Scientific Multiskan FC plate reader. Growth curves were determined by light scattering at 620 nm. The BactEXTRACT app was used to perform analysis and visualization of the data (*Dénéréaz and Veening, 2024*).

## Protein purification

### Purification of *bsuS*mc and biotinylated *bsuS*mc(C119S, C437S, C826S, E1118Q, R643C)

*bsuS*mc proteins were purified according to *Bürmann et al., 2017*. pET-22 or pET-28 plasmids encoding the *Smc* recombinant sequences were transformed into *E. coli* BL21-Gold (DE3) cells. Protein expression was carried out in ZYM-5052 autoinduction medium for 23 hours at 24°C. The cells were harvested and resuspended in lysis buffer (50 mM Tris-HCl, pH 7.5, 150 mM NaCl, 1 mM EDTA, 1 mM DTT, 10% [w/v] sucrose) supplemented with protease inhibitor cocktail. Cell lysis was achieved by sonication, and the lysate was clarified by centrifugation. The supernatant was filtered through a 0.45 μm membrane and loaded onto two HiTrap Blue HP 5 mL columns connected in series. Elution was performed using lysis buffer with 1 M NaCl. The main peak fractions were pooled and diluted with salt-free buffer (50 mM Tris-HCl, pH 7.5; 1 mM EDTA; 1 mM DTT) to a final conductivity equivalent to 50 mM NaCl (~8 mS/cm). This diluted sample, supplemented with protease inhibitor cocktail, was

applied to a HiTrap Heparin HP 5 mL column, and the proteins were eluted using a linear gradient up to 2 M NaCl. The main peak fractions (~5 mL) were collected and subjected to further purification by gel filtration using an XK 16/70 Superose 6 PG column equilibrated with 50 mM Tris-HCl, pH 7.5, 200 mM NaCl, 1 mM EDTA, and 1 mM TCEP. The peak fractions were collected, concentrated using a Vivaspin 15 10K MWCO filter, flash frozen in liquid nitrogen, and stored at –80°C. Protein concentration was determined by absorbance, utilizing theoretical molar absorption and molecular weight values.

For the biotinylated *bsu*Smc(C119S, C437S, C826S, E1118Q, R643C) protein used in the sybody selection, the protein was first purified as described above, except that the reducing agent (TCEP) was omitted from the final gel filtration buffer. Labeling was performed by incubating 600 µL of *bsu*Smc(C119S, C437S, C826S, E1118Q, R643C) (83 µM in 50 mM Tris-HCl pH 7.5, 200 mM NaCl) with 1 mM PEG2-biotin maleimide for 10 minutes at 4°C. The reaction was quenched by addition of 0.5 mM 2-mercaptoethanol. Excess label was removed using Zeba spin desalting columns (Thermo Fisher) in multiple parallel runs due to volume limitations. The final yield was 600 µL at 72.35 µM (subunit concentration). Labeled *bsu*Smc(C119S, C437S, C826S, E1118Q, R643C) was loaded onto a Superose 6 10/300 Increase column pre-equilibrated in 50 mM Tris-HCl pH 7.5, 200 mM NaCl. Peak fractions were pooled and mixed with ScpA and ScpB at a 1:1:2 molar ratio relative to the *bsu*Smc dimer. The final concentration of the full complex was adjusted to 5.7 µM. Complexes were aliquoted in 100 µL portions, flash-frozen in liquid nitrogen, and stored at –80°C.

## Purification of *bsu*ScpA

ScpA was purified using the method described by *Vazquez Nunez et al., 2019*. *E. coli* BL21-Gold (DE3) cells, transformed with a pET-28 derived plasmid encoding the ScpA protein, were used for expression. Cultivation was performed in ZYM-5052 autoinduction medium at 16°C for 28 hours. Cells were then harvested and resuspended in lysis buffer (50 mM Tris-HCl, pH 7.5; 200 mM NaCl; 5% glycerol) supplemented with protease inhibitor cocktail. The cells were lysed by sonication, and the lysate was clarified by centrifugation. The supernatant was applied to a 5 mL HiTrap Q ion exchange column and eluted with a gradient up to 2 M NaCl. Peak fractions were pooled and adjusted to a final concentration of 3 M NaCl by mixing with 4 M NaCl buffer. This mixture was loaded onto a HiTrap Butyl HP column and eluted with a reverse gradient to 50 mM NaCl. Eluted peak fractions were concentrated to 5 mL using Vivaspin 15 10K MWCO filters and further purified by size-exclusion chromatography (SEC) on a HiLoad 16/600 Superdex 75 pg column equilibrated with 20 mM Tris-HCl, pH 7.5, and 200 mM NaCl. The purified protein was concentrated, flash frozen, and stored at –80°C.

## Purification of *bsu*ScpB

ScpB was purified following the protocol outlined by *Vazquez Nunez et al., 2019*. The coding sequence of ScpB, cloned into a pET-22 derived plasmid, was transformed into chemically competent BL21-Gold (DE3) *E. coli* cells. These cells were cultivated in ZYM-5052 autoinduction medium at 24°C for 23 hours. Subsequently, the cells were harvested and resuspended in lysis buffer (50 mM Tris-HCl, pH 7.5; 150 mM NaCl; 1 mM EDTA; 1 mM DTT) supplemented with protease inhibitor cocktail. Cell lysis was performed by sonication, followed by centrifugation to remove cell debris. The resulting supernatant was diluted to a final NaCl concentration of 50 mM and loaded onto a 5 mL HiTrap Q HP column. Elution was achieved using a gradient up to 2 M NaCl. The eluate was then diluted with lysis buffer containing 4 M NaCl to achieve a final concentration of 3 M NaCl. This sample was applied to two 5 mL HiTrap Butyl columns connected in series, and the protein was eluted with a reverse gradient to 50 mM NaCl. The peak fractions from this column were concentrated and subjected to SEC using a HiLoad 16/600 Superdex 200 pg column equilibrated with 50 mM Tris-HCl, pH 7.5, 100 mM NaCl, and 1 mM DTT. Fractions containing ScpB were concentrated, flash frozen, and stored at –80°C.

## Medium-scale sybody purification

Individual sybody plasmids were transformed into chemically competent *E. coli* MC1061 cells via heat shock at 42°C for 45 seconds, followed by recovery in LB medium at 37°C. Transformed cells were cultured overnight in TB medium supplemented with 25 µg/mL chloramphenicol. Precultures were used to inoculate 50 mL TB cultures, which were grown at 37°C before shifting to 22°C. Expression was induced with 0.02% (wt/vol) L-(+)-arabinose and continued overnight at 22°C with shaking. Cells

were harvested by centrifugation and resuspended in periplasmic extraction buffer. Following incubation at 4°C, cells were pelleted, and the supernatant was supplemented with imidazole to 15 mM. The extract was incubated with His MultiTrap HP resin, followed by centrifugation and washing with TBS containing 30 mM imidazole. Elution was performed using TBS with 300 mM imidazole. Purified sybodies were analyzed on a Sepax SRT-10C SEC100 column at 1 mL/minute flow rate. Monomeric sybodies eluted between 11–12.5 mL, while retention volumes <11 mL indicated oligomerization, and >14 mL suggested strong column interaction. Non-expressed, oligomeric, or highly interacting sybodies were discarded. Typical yields ranged from 200 µg to 1 mg (*Zimmermann et al., 2020*).

## Fluorescence imaging of ParB-GFP strains

### Image acquisition

*B. subtilis* cells were first cultured in LB supplemented with 5 µg/mL chloramphenicol and 0.5% glucose at 30°C, from which a day culture was inoculated to 0.005 in LB with 5 µg/mL chloramphenicol. When needed, cultures were induced at $OD_{600}$ ~0.02 with 0.5% xylose and grown an extra 30 minutes until imaging at an $OD_{600}$ of 0.04. For microscopy analysis, 0.5 µL of the cell suspension was spotted onto agarose-coated microscopy slides. Images were acquired using a Leica DMi8 microscope equipped with an sCMOS DFC9000 (Leica) camera, a SOLA light engine (Lumencor), and a 100×/1.40 oil-immersion objective. Exposure time for image acquisition was set to 600ms. Acquired images were processed using LAS X Office software (v.1.4.7.28982, Leica Microsystems).

### Image analysis

Microscopy images were analyzed using a fully automated image processing pipeline. Cell segmentation was performed with the pretrained Omnipose model (*Cutler et al., 2022*) implemented in Cellpose (*Stringer et al., 2021*), which is optimized for bacterial morphologies. Prior to segmentation, a normalization step was applied to standardize image contrast. Following segmentation, intracellular foci were detected using Spotiflow (*Dominguez Mantes et al., 2025*), a dedicated model trained for spot identification. Each detected spot was assigned to an individual cell based on spatial location, enabling quantification of the number of foci per cell. Cell morphology analysis was also performed to estimate cell length.

Quantitative results were compiled into a structured dataset, including the number of cells per image, percentage of cells lacking foci, number of foci per cell, cell length, and number of foci per micron. Data visualization and statistical analysis were performed using GraphPad Prism 10.4.1 (627).

## ELISA

*E. coli* MC1061 colonies were screened for sybody expression, and periplasmic extracts were obtained as per the original protocol (*Zimmermann et al., 2020*). ELISA plates were coated with Protein A and anti-c-Myc antibody before incubation with sybody-containing extracts. Three wells per sybody were incubated with biotinylated maltose binding protein (MBP, negative control) or with 50 nM biotinylated bsSmc-ScpAB(E1118Q) with ATP/dsDNA or without. Washes and ELISA development were performed using TBSM-D. Absorbance was measured at 650 nm.

## ATPase measurements

ATPase activity was assessed using the coupled pyruvate kinase/lactate dehydrogenase reaction as described by *Bürmann et al., 2017*. ADP production was monitored over a period of 1 hour by measuring the absorbance changes of NADH at 340 nm. Data collection was performed using a Synergy Neo Hybrid Multi-Mode Microplate Reader. The reaction mixture comprised 1 mM NADH, 3 mM phosphoenolpyruvic acid, 100 U pyruvate kinase, 20 U lactate dehydrogenase, and varying concentrations of ATP. For assays requiring double-stranded oligonucleotides, a 40 bp oligonucleotide (5'-TTAGTTGTTC GTAGTGCTCG TCTGGCTCTG GATTACCCGC-3') was added to a final concentration of 3 mM. The final protein concentration in the assay was 0.15 µM *bsu*Smc dimers in ATPase assay buffer (50 mM HEPES-KOH, pH 7.5; 50 mM NaCl; 2 mM $MgCl_2$). All measurements were conducted at 25°C.

## Acknowledgements

We are grateful to Roberto Vazquez Nunez for the biotinylated bsuSmc(C119S, C437S, C826S, E1118Q, R643C) protein preparation, Arianna Ravera from the DCSR computing facility for help in image analysis and data extraction of the fluorescence microscopy images. Members of the Gruber Lab for critical feedback of the manuscript and for stimulating discussions. This study was supported by internal funding, Swiss National Science Foundation (310030_215138, to M.A.S) and by the European Research Council (Horizon 2020 ERC CoG 724482, to S.G.).

## Additional information

### Funding

| Funder | Grant reference number | Author |
|---|---|---|
| HORIZON EUROPE European Research Council | 724482 | Stephan Gruber |
| Swiss National Science Foundation | 310030_215138 | Markus A Seeger |

The funders had no role in study design, data collection and interpretation, or the decision to submit the work for publication.

### Author contributions

Ophélie J Gosselin, Investigation, Methodology, Writing – original draft, Writing – review and editing; Michael Taschner, Investigation, Methodology, Writing – review and editing; Lea M Huber-Hürlimann, Methodology; Markus A Seeger, Conceptualization, Supervision, Investigation, Methodology, Writing – review and editing; Stephan Gruber, Conceptualization, Supervision, Funding acquisition, Investigation, Methodology, Writing – original draft, Writing – review and editing

### Author ORCIDs

Ophélie J Gosselin ⓘ https://orcid.org/0009-0004-8996-4689
Michael Taschner ⓘ https://orcid.org/0000-0002-8881-3705
Markus A Seeger ⓘ https://orcid.org/0000-0003-1761-8571
Stephan Gruber ⓘ https://orcid.org/0000-0002-0150-0395

Reviewer #1 (Public review): https://doi.org/10.7554/eLife.111131.3.sa1
Reviewer #2 (Public review): https://doi.org/10.7554/eLife.111131.3.sa2
Reviewer #3 (Public review): https://doi.org/10.7554/eLife.111131.3.sa3
Author response https://doi.org/10.7554/eLife.111131.3.sa4

## Additional files

### Supplementary files

Supplementary file 1. ELISA sybody binding assay.

Supplementary file 2. Sybody sequences.

Supplementary file 3. Percentage of cells lacking ParB-GFP foci, indicating the absence of chromosome due to segregation defect. In presence and absence of xylose for sybody induction.

Supplementary file 4. Strains, Plasmids, Oligonucleotides.

MDAR checklist

### Data availability

All data generated or analyzed during this study are included in the manuscript, its figures, figure supplements, and source data files, or are available deposited in Mendeley Data under DOI: https://doi.org/10.17632/k6k62p7z2s.1. Newly generated materials are available from the corresponding author upon reasonable request.

The following dataset was generated:

| Author(s) | Year | Dataset title | Dataset URL | Database and Identifier |
| --- | --- | --- | --- | --- |
| Gosselin O, Taschner M, Huber-Hürlimann L, Seeger M, Gruber S | 2025 | Single Domain Antibody Inhibitors Target the Coiled Coil Arms of the *Bacillus subtilis* SMC complex | https://doi.org/10.17632/k6k62p7z2s.1 | Mendeley Data, 10.17632/k6k62p7z2s.1 |

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
