## [Editor Report · eLife Assessment]

This **important** study introduces an innovative synthetic nanobody approach to probe the function of the bacterial SMC complex. The work is a **compelling** example of the potential of this approach. The authors generate protein chimeras to provide **convincing** evidence that their identified nanobodies target the coiled-coil region of the SMC subunit, demonstrating that this region is critical for SMC function in vivo. Overall, the work is significant for the fields of genome organization, SMC protein biology, synthetic biology, and bacterial cell biology.

[Editors' note: this paper was reviewed by Review Commons.]

---

## [Referee Report · Reviewer #1 (Public review)]

[Editors' note: this version has been assessed by the Reviewing Editor without further input from the original reviewers. The authors have addressed the major comments raised in the previous round of review. Public Reviews below refer to the version submitted to Review Commons.]

Summary:

Gosselin et al., develop a method to target protein activity using synthetic single-domain nanobodies (sybodies). They screen a library of sybodies using ribosome/ phage display generated against bacillus Smc-ScpAB complex. Specifically, they use an ATP hydrolysis deficient mutant of SMC so as to identify sybodies that will potentially disrupt Smc-ScpAB activity. They next screen their library in vivo, using growth defects in rich media as a read-out for Smc activity perturbation. They identify 14 sybodies that mirror smc deletion phenotype including defective growth in fast-growth conditions, as well as chromosome segregation defects. The authors use a clever approach by making chimeras between bacillus and S. pnuemoniae Smc to narrow-down to specific regions within the bacillus Smc coiled-coil that are likely targets of the sybodies. Using ATPase assays, they find that the sybodies either impede DNA-stimulated ATP hydrolysis or hyperactivate ATP hydrolysis (even in the absence of DNA). The authors propose that the sybodies may likely be locking Smc-ScpAB in the "closed" or "open" state via interaction with the specific coiled-coil region on Smc. I have a few comments that the authors should consider:

Major comments:

(1) Lack of direct in vitro binding measurements:

The authors do not provide measurements of sybody affinities, binding/ unbinding kinetics, stoichiometries with respect to Smc-ScpAB. Additionally, do the sybodies preferentially interact with Smc in ATP/ DNA-bound state? And do the sybodies affect the interaction of ScpAB with SMC?

It is understandable that such measurements for 14 sybodies is challenging, and not essential for this study. Nonetheless, it is informative to have biochemical characterization of sybody interaction with the Smc-ScpAB complex for at least 1-2 candidate sybodies described here.

(2) Many modes of sybody binding to Smc are plausible

The authors provide an elaborate discussion of sybodies locking the Smc-ScpAB complex in open/ closed states. However, in the absence of structural support, the mechanistic inferences may need to be tempered. For example, is it also not possible for the sybodies to bind the inner interface of the coiled-coil, resulting in steric hinderance to coiled-coil interactions. It is also possible that sybody interaction disrupts ScpAB interaction (as data ruling this possibility out has not been provided). Thus, other potential mechanisms would be worth considering/ discussing. In this direction, did AlphaFold reveal any potential insights into putative binding locations?

(3) Sybody expression in vivo

Have the authors estimated sybody expression in vivo? Are they all expressed to similar levels?

(4) Sybodies should phenocopy ATP hydrolysis mutant of Smc

The sybodies were screened against an ATP hydrolysis deficient mutant of Smc, with the rationale that these sybodies would interfere this step of the Smc duty cycle. Does the expression of the sybodies in vivo phenocopy the ATP hydrolysis deficient mutant of Smc? Could the authors consider any phenotypic read-outs that can indicate whether the sybody action results in an smc-null effect or specifically an ATP hydrolysis deficient effect?

Significance:

Overall, this is an impressive study that uses an elegant strategy to find inhibitors of protein activity in vivo. The manuscript is clearly written and the experiments are logical and well-designed. The findings from the study will be significant to the broad field of genome biology, synthetic biology and also SMC biology. Specifically, the coiled coil domain of SMC proteins has been proposed to be of high functional value. The authors have elegantly identified key coiled-coil regions that may be important for function, and parallelly exhibited potential of the use of synthetic sybody/designed binders for inhibition of protein activity.

---

## [Referee Report · Reviewer #2 (Public review)]

Summary:

Structural Maintenance of Chromosome proteins (SMCs), a family of proteins found in almost all organisms, are organizers of DNA. They accomplish this by a process known as loop extrusion, wherein double-stranded DNA is actively reeled in and extruded into loops. Although SMCs are known to have several DNA binding regions, the exact mechanism by which they facilitate loop extrusion is not understood but is believed to entail large conformational changes. There are currently several models for loop extrusion, including one wherein the coiled coil (CC) arms open, but there is a lack of insightful experimentation and analysis to confirm any of these models. The work presented aims to provide much-needed new tools to investigate these questions: conformation-selective sybodies (synthetic nanobodies) that are likely to alter the CC opening and closing reactions.

The authors produced, isolated, and expressed sybodies that specifically bound to *Bacillus subtilis* Smc-ScpAB. Using chimeric Smc constructs, where the coiled coils were partly replaced with the corresponding sequences from Streptococcus pneumoniae, the authors revealed that the isolated sybodies all targeted the same 4N CC element of the Smc arms. This region is likely disrupted by the sybodies either by stopping the arms from opening (correctly) or forcing them to stay open (enough). Disrupting these functional elements is suggested to cause the Smc-dependent chromosome organization lethal phenotype, implying that arm opening and closing is a key regulatory feature of bacterial Smc-ScpAB.

Significance:

The authors present a new method for trapping bacterial Smc's in certain conformations using synthetic antibodies. Using these antibodies, they have pinpointed the (previously suggested) 4N region of the coiled coils as an essential site for the opening and closing of the Smc coiled coil arms and that hindering these reactions blocks Smc-driven chromosomal organization. The work has important implications for how we might elucidate the mechanism of DNA loop extrusion by SMC complexes.

---

## [Referee Report · Reviewer #3 (Public review)]

Summary:

Gosselin et al. use the sybody technology to study effects of in vivo inhibition of the *Bacillus subtilis* SMC complex. Smc proteins are central DNA binding elements of several complexes that are vital for chromosome dynamics in almost all organisms. Sybodies are selected from three different libraries of the single domain antibodies, using the "transition state" mutant Smc. They identify 14 such mutant sybodies that are lethal when expressed in vivo, because they prevent proper function of Smc. The authors present evidence suggesting that all obtained sybodies bind to a coiled-coil region close to the Smc "neck", and thereby interfere with the Smc activity cycle, as evidenced by defective ATPase activity when Smc is bound to DNA.

The study is well done and presented and shows that the strategy is very potent in finding a means to quickly turn off a protein's function in vivo, much quicker than depleting the protein.

The authors also draw conclusions on the molecular mode of action of the SMC complex. The provide a number of suggestive experiments, but in my view mostly indirect evidence for such mechanism.

My main criticism is that the authors have used a single - and catalytically trapped form of SMC. They speculate why they only obtain sybodies from one library, and then only identify sybodies that bind to a rather small part of the large Smc protein. While the approach is definitely valuable, it is biassed towards sybodies that bind to Smc in a quite special way, it seems. Using wild type Smc would be interesting, to make more robust statements about the action of sybodies potentially binding to different parts of Smc.

Line 105: Alternatively, the other libraries did not produce good binders or these sybodies were 106 not stably expressed in *B. subtilis*. This could be tested using Western blotting - I am assuming sybody antibodies are commercially available. However, this test is not important for the overall study, it would just clarify a minor point.

Fig. 2B: is odd to count Spo0J foci per cells, as it is clear from the images that several origins must be present within the fluorescent foci. I am fine with the "counting" method, as the images show there is a clear segregation defect when sybodies are expressed, I believe the authors should state, though, that this is not a replication block, but failure to segregate origins.

Testing binding sites of sybodies to the SMC complex is done in an indirect manner, by using chimeric Smc constructs. I am surprised why the authors have not used in vitro crosslinking: the authors can purify Smc, and mass spectrometry analyses would identify sites where sybodies are crosslinked to Smc. Again, I am fine with the indirect method, but the authors make quite concrete statements on binding based on non-inhibition of chimeric Smc; I can see alternative explanations why a chimera may not be targeted.

Smc-disrupting sybodies affect the ATPase activity in one of two ways. Again, rather indirect experiments. This leads to the point Revealing Smc arm dynamics through synthetic binders in the discussion. The authors are quite careful in stating that their experiments are suggestive for a certain mode of action of Smc, which is warranted.

In line 245, they state More broadly, the study demonstrates how synthetic binders can trap, stabilize, or block transient conformations of active chromatin-associated machines, providing a powerful means to probe their mechanisms in living cells. This is off course a possible scenario for the use of sybodies, but the study does not really trap Smc in a transient conformation, at least this is not clearly shown.

Overall, it is an interesting study, with a well-presented novel technology, and a limited gain of knowledge on SMC proteins.

Significance:

The work describes the gaining and use of single-binder antibodies (sybodies) to interfere with the function of proteins in bacteria. Using this technology for the SMC complex, the authors demonstrate that they can obtain a significant of binders that target a defined region is SMC and thereby interfere with the ATPase cycle.

The study does not present a strong gain of knowledge of the mode of action of the SMC complex.

---

## [Author Response]

The following is the authors’ response to the original reviews

**Public Reviews:**

**Reviewer #1 (Public review):**
Summary:Gosselin et al., develop a method to target protein activity using synthetic single-domain nanobodies (sybodies). They screen a library of sybodies using ribosome/ phage display generated against bacillus Smc-ScpAB complex. Specifically, they use an ATP hydrolysis deficient mutant of SMC so as to identify sybodies that will potentially disrupt Smc-ScpAB activity. They next screen their library in vivo, using growth defects in rich media as a read-out for Smc activity perturbation. They identify 14 sybodies that mirror smc deletion phenotype including defective growth in fast-growth conditions, as well as chromosome segregation defects. The authors use a clever approach by making chimeras between bacillus and S. pnuemoniae Smc to narrow-down to specific regions within the bacillus Smc coiled-coil that are likely targets of the sybodies. Using ATPase assays, they find that the sybodies either impede DNA-stimulated ATP hydrolysis or hyperactivate ATP hydrolysis (even in the absence of DNA). The authors propose that the sybodies may likely be locking Smc-ScpAB in the "closed" or "open" state via interaction with the specific coiled-coil region on Smc. I have a few comments that the authors should consider:Major comments:(1) Lack of direct in vitro binding measurements:The authors do not provide measurements of sybody affinities, binding/ unbinding kinetics, stoichiometries with respect to Smc-ScpAB. Additionally, do the sybodies preferentially interact with Smc in ATP/ DNA-bound state? And do the sybodies affect the interaction of ScpAB with SMC?It is understandable that such measurements for 14 sybodies is challenging, and not essential for this study. Nonetheless, it is informative to have biochemical characterization of sybody interaction with the Smc-ScpAB complex for at least 1-2 candidate sybodies described here.

We agree with the reviewer that adding such data would be reassuring and that obtaining solid data using purified components is not trivial, even for a smaller selection of sybodies. We have now incorporated ELISA data as new Table S1, which shows that most sybodies support clear binding to Smc-ScpAB. Curiously, while (only) some sybodies show a clear preference for ATP-bound or unbound Smc, this is not a strong predictor of the strength of phenotype observed in vivo. We have also attempted to characterize the binding of Smc to sybodies by other methods including pull-downs, cross-linking, and by biophysical methods (GCI). However, we prefer to not include these data as the outcomes are not clear due to inconsistencies in the behaviour of purified sybodies.

(2) Many modes of sybody binding to Smc are plausibleThe authors provide an elaborate discussion of sybodies locking the Smc-ScpAB complex in open/ closed states. However, in the absence of structural support, the mechanistic inferences may need to be tempered. For example, is it also not possible for the sybodies to bind the inner interface of the coiled-coil, resulting in steric hinderance to coiled-coil interactions. It is also possible that sybody interaction disrupts ScpAB interaction (as data ruling this possibility out has not been provided). Thus, other potential mechanisms would be worth considering/ discussing. In this direction, did AlphaFold reveal any potential insights into putative binding locations?

We have attempted to map the binding by structure prediction, however, so far, even the latest versions of AlphaFold are not able to clearly delineate the binding interface that we have confidently identified by the mapping using chimeric proteins. Indeed, many ways of binding are possible, including disruption of ScpAB interaction. However, since the mapped binding sites are located on the SMC coiled coils, the later scenario seems unlikely and would be an indirect consequence of altered coiled coil configuration, consistent with our current interpretation.

(3) Sybody expression in vivoHave the authors estimated sybody expression in vivo? Are they all expressed to similar levels?

We have tagged selected sybodies with gfp and performed live cell imaging. This shows that sybodies without strong phenotypes are similarly expressed at least at low inducer concentration. Moreover, many sybodies localize as foci in the cell presumably by binding to Smc complexes loaded onto the chromosome at ParB/parS sites. We have included example data in the revised version of the manuscript as Figure S4 and Figure S5. Notably, a sybody (Sb007) with a weak growth phenotype shows focal localization at low inducer concentration and high expression levels when fully induced, comparable to sybodies with strong phenotypes. Altogether, this suggests that the lack of phenotype is not due to absence of sybody expression or localization.

(4) Sybodies should phenocopy ATP hydrolysis mutant of SmcThe sybodies were screened against an ATP hydrolysis deficient mutant of Smc, with the rationale that these sybodies would interfere this step of the Smc duty cycle. Does the expression of the sybodies in vivo phenocopy the ATP hydrolysis deficient mutant of Smc? Could the authors consider any phenotypic read-outs that can indicate whether the sybody action results in an smc-null effect or specifically an ATP hydrolysis deficient effect?

As alluded to above, we think that our selection gave rise to sybodies that bind various, possibly multiple Smc conformations. Consistent with this idea, the phenotypes of sybody expression are similar to null mutant rather than the ATP-hydrolysis defective EQ mutant, which display even more severe growth phenotypes in *B. subtilis*. To highlight this point, we have added the following notes to the text:

“These conditions favour ATP-engaged particles alongside the typically predominant ATP-disengaged rod-shaped state.”

“ELISA data revealed that nearly all clones bind purified Smc-ScpAB (Table 1). However, the ELISA signals of only few Sybodies showed clear dependence on the presence or absence of ATP and DNA (Table S1).”

Significance:Overall, this is an impressive study that uses an elegant strategy to find inhibitors of protein activity in vivo. The manuscript is clearly written and the experiments are logical and well-designed. The findings from the study will be significant to the broad field of genome biology, synthetic biology and also SMC biology. Specifically, the coiled coil domain of SMC proteins have been proposed to be of high functional value. The authors have elegantly identified key coiled-coil regions that may be important for function, and parallelly exhibited potential of the use of synthetic sybody/designed binders for inhibition of protein activity.
**Reviewer #2 (Public review):**
Summary:Structural Maintenance of Chromosome proteins (SMCs), a family of proteins found in almost all organisms, are organizers of DNA. They accomplish this by a process known as loop extrusion, wherein double-stranded DNA is actively reeled in and extruded into loops. Although SMCs are known to have several DNA binding regions, the exact mechanism by which they facilitate loop extrusion is not understood but is believed to entail large conformational changes. There are currently several models for loop extrusion, including one wherein the coiled coil (CC) arms open, but there is a lack of insightful experimentation and analysis to confirm any of these models. The work presented aims to provide much-needed new tools to investigate these questions: conformation-selective sybodies (synthetic nanobodies) that are likely to alter the CC opening and closing reactions.The authors produced, isolated, and expressed sybodies that specifically bound to *Bacillus subtilis* Smc-ScpAB. Using chimeric Smc constructs, where the coiled coils were partly replaced with the corresponding sequences from Streptococcus pneumoniae, the authors revealed that the isolated sybodies all targeted the same 4N CC element of the Smc arms. This region is likely disrupted by the sybodies either by stopping the arms from opening (correctly) or forcing them to stay open (enough). Disrupting these functional elements is suggested to cause the Smc-dependent chromosome organization lethal phenotype, implying that arm opening and closing is a key regulatory feature of bacterial Smc-ScpAB.Significance:The authors present a new method for trapping bacterial Smc's in certain conformations using synthetic antibodies. Using these antibodies, they have pinpointed the (previously suggested) 4N region of the coiled coils as an essential site for the opening and closing of the Smc coiled coil arms and that hindering these reactions blocks Smc-driven chromosomal organization. The work has important implications for how we might elucidate the mechanism of DNA loop extrusion by SMC complexes.
**Reviewer #3 (Public review):**
Summary:Gosselin et al. use the sybody technology to study effects of in vivo inhibition of the *Bacillus subtilis* SMC complex. Smc proteins are central DNA binding elements of several complexes that are vital for chromosome dynamics in almost all organisms. Sybodies are selected from three different libraries of the single domain antibodies, using the "transition state" mutant Smc. They identify 14 such mutant sybodies that are lethal when expressed in vivo, because they prevent proper function of Smc. The authors present evidence suggesting that all obtained sybodies bind to a coiled-coil region close to the Smc "neck", and thereby interfere with the Smc activity cycle, as evidenced by defective ATPase activity when Smc is bound to DNA.The study is well done and presented and shows that the strategy is very potent in finding a means to quickly turn off a protein's function in vivo, much quicker than depleting the protein.The authors also draw conclusions on the molecular mode of action of the SMC complex. The provide a number of suggestive experiments, but in my view mostly indirect evidence for such mechanism.My main criticism is that the authors have used a single - and catalytically trapped form of SMC. They speculate why they only obtain sybodies from one library, and then only identify sybodies that bind to a rather small part of the large Smc protein. While the approach is definitely valuable, it is biassed towards sybodies that bind to Smc in a quite special way, it seems. Using wild type Smc would be interesting, to make more robust statements about the action of sybodies potentially binding to different parts of Smc.

The reviewer reports (Rev. #1 and Rev. #3) made us realize that the manuscript text was misleading on the this point. Although we used the purified ATP hydrolysis–deficient Smc protein for sybody isolation, this is not expected to restrict the selection to a specific conformation. As described in detail in Vazquez-Nunez et al. (Figure 5), this mutant displays the ATP-engaged conformation only in a smaller fraction of complexes (~25% in the presence of ATP and DNA), consistent with prior in vivo observations reported by Diebold-Durand et al. (Figure 5). Rather than limiting the selection to a particular configuration, our aim was to reduce the prevalence of the predominant rod state in order to broaden the range of conformations represented during sybody selection. Consistent with this interpretation, only a small number of isolated sybodies show strong conformation-specific binding in the presence or absence of ATP/DNA, as observed by ELISA (now included in the manuscript). Notably, the effect size of ATP/DNA on ELISA signals was not a strong predictor to the strength of phenotypes observed in vivo. The text has been revised accordingly. See line 84 and line 92.

We are thus quite confident based prior work (and on the now included ELISA data) that the Smc ATPase mutation did not strongly bias the selection in one way or another. The surprising bias towards coiled coil binding sites has likely other explanations, as they likely form a preferred epitope recognized by sybodies from the loop library.

Line 105: Alternatively, the other libraries did not produce good binders or these sybodies were 106 not stably expressed in *B. subtilis*. This could be tested using Western blotting - I am assuming sybody antibodies are commercially available. However, this test is not important for the overall study, it would just clarify a minor point.

While there are antibody fragments available to augment the size of sybodies (PMID: 40108246), these recognize 3D-epitopes and are thus not suited for Western blotting. We did not follow up on the negative results of two of the three libraries but would like to point out again that there are several biases that likely emerge for the same reason (bias to library, bias to coiled coil binding site). If correct, then sybodies are likely ineffective in inactivating Smc in *B. subtilis*, with the notable exceptions of the sybodies that we have isolated and characterized in this manuscript. We have added this notion to the manuscript.

Fig. 2B: is odd to count Spo0J foci per cells, as it is clear from the images that several origins must be present within the fluorescent foci. I am fine with the "counting" method, as the images show there is a clear segregation defect when sybodies are expressed, I believe the authors should state, though, that this is not a replication block, but failure to segregate origins.

We agree that this is an important point. We have added the following statement to clarify this point: “These elongated cells are known to harbour expanded nucleoids, consistent with delayed oriC separation rather than delayed DNA replication”

Testing binding sites of sybodies to the SMC complex is done in an indirect manner, by using chimeric Smc constructs. I am surprised why the authors have not used in vitro crosslinking: the authors can purify Smc, and mass spectrometry analyses would identify sites where sybodies are crosslinked to Smc. Again, I am fine with the indirect method, but the authors make quite concrete statements on binding based on non-inhibition of chimeric Smc; I can see alternative explanations why a chimera may not be targeted.

We have made several attempts of testing direct binding with mixed outcomes and decided to not include those results in the light of the stronger and more relevant in vivo mapping. However, we have added ELISA results (new Table S1) that support a direct interaction.

Smc-disrupting sybodies affect the ATPase activity in one of two ways. Again, rather indirect experiments. This leads to the point Revealing Smc arm dynamics through synthetic binders in the discussion. The authors are quite careful in stating that their experiments are suggestive for a certain mode of action of Smc, which is warranted.In line 245, they state More broadly, the study demonstrates how synthetic binders can trap, stabilize, or block transient conformations of active chromatin-associated machines, providing a powerful means to probe their mechanisms in living cells. This is off course a possible scenario for the use of sybodies, but the study does not really trap Smc in a transient conformation, at least this is not clearly shown.

We agree and have simplified the statement by removing “stabilize” and “transient”.

Overall, it is an interesting study, with a well-presented novel technology, and a limited gain of knowledge on SMC proteins.

We respectfully disagree with the last point, since our unique results highlight the importance of the Smc coiled coils. which are less well represented in the SMC literature (when compared to the heads and hinge domains for example), likely (at least in part) due the mild effect of single point mutations on coiled coil dynamics.

Significance:The work describes the gaining and use of single-binder antibodies (sybodies) to interfere with the function of proteins in bacteria. Using this technology for the SMC complex, the authors demonstrate that they can obtain a significant of binders that target a defined region is SMC and thereby interfere with the ATPase cycle.The study does not present a strong gain of knowledge of the mode of action of the SMC complex.

As pointed out above, we respectfully disagree with this assertion.